# Drought Stress Responses in Context-Specific Genome-Scale Metabolic Models of *Arabidopsis thaliana*

**DOI:** 10.3390/metabo10040159

**Published:** 2020-04-18

**Authors:** Ratklao Siriwach, Fumio Matsuda, Kentaro Yano, Masami Yokota Hirai

**Affiliations:** 1RIKEN Center for Sustainable Resource Science, Yokohama, Kanagawa 230-0045, Japan; ratklao.siriwach@riken.jp; 2Department of Bioinformatic Engineering, Graduate School of Information Science and Technology, Osaka University, Suita, Osaka 565-0871, Japan; fmatsuda@bio.eng.osaka-u.ac.jp; 3Bioinformatics Laboratory, Department of Life Sciences, School of Agriculture, Meiji University, Kawasaki, Kanagawa 214-8571, Japan; kyano@meiji.ac.jp

**Keywords:** *Arabidopsis*, drought, flux balance analysis, genome-scale metabolic model, metabolism, metabolome, transcriptome

## Abstract

Drought perturbs metabolism in plants and limits their growth. Because drought stress on crops affects their yields, understanding the complex adaptation mechanisms evolved by plants against drought will facilitate the development of drought-tolerant crops for agricultural use. In this study, we examined the metabolic pathways of *Arabidopsis thaliana* which respond to drought stress by omics-based in silico analyses. We proposed an analysis pipeline to understand metabolism under specific conditions based on a genome-scale metabolic model (GEM). Context-specific GEMs under drought and well-watered control conditions were reconstructed using transcriptome data and examined using metabolome data. The metabolic fluxes throughout the metabolic network were estimated by flux balance analysis using the context-specific GEMs. We used in silico methods to identify an important reaction contributing to biomass production and clarified metabolic reaction responses under drought stress by comparative analysis between drought and control conditions. This proposed pipeline can be applied in other studies to understand metabolic changes under specific conditions using *Arabidopsis* GEM or other available plant GEMs.

## 1. Introduction

In the era of global climate change and increasing food demand caused by population increases, it may become challenging to grow enough agricultural crops to produce sufficient amounts of foods for human consumption. Drought is among the major environmental problems in agriculture worldwide, as water deficits limit plant growth and reduce crop production [1,2]. Therefore, understanding how plants cope with drought has become a major research focus in plant science.

Plants have evolved phenotypic plasticity under changing environments and have altered their metabolism to balance their growth and specific stress responses. Plants utilize various strategies to cope with drought stress of different degrees and durations. The complex regulatory circuits under drought stress lead to physiological or morphological changes across a range of temporal and spatial scales. One well-known regulatory circuit is the abscisic acid (ABA) biosynthesis and signaling pathway. When water deficit occurs, plant cells undergo osmotic changes and induce regulatory genes. Subsequently, ABA levels are transiently increased [3,4]. ABA triggers the expression of many drought-stress-associated genes, resulting in the accumulation of protective proteins, increasing the levels of compatible solutes (e.g., sugars and proline) and antioxidants (e.g., flavonoids and polyphenols), and finally suppressing energy-consuming pathways [5,6,7]. Regulatory elements, such as transcription factors and protein kinases, have been intensively studied in the molecular genetics field. However, metabolic adaptation, which is also important for the cell to maintain a steady state (homeostasis), has not been widely examined. Considering the intricate mechanism of drought stress responses, it is difficult to observe the entire metabolic adaptation process at a specific temporal and spatial point using current technologies. Additionally, as eukaryotes contain organelles, the distribution and localization of metabolites in plants are difficult to examine at the subcellular level.

To understand the whole metabolic system response, simulation using biochemical metabolic networks, known as a genome-scale metabolic model (GEM), is performed [8]. Based on the concept of system biology, simulation of the metabolite distribution across diverse conditions can be performed to estimate and interpret stress-related metabolic adaptation [8,9]. GEM is a metabolic network of chemical reactions constructed from genome sequence information and detailed pathway information from the literature [10]. In a GEM, metabolic flux (i.e., producing-consuming flow rates of metabolites through a metabolic reaction) can be assessed by several mathematics computational approaches. Among these approaches, flux balance analysis (FBA) is commonly used [11]. The core feature of FBA is metabolic reactions, which are represented as a stoichiometric matrix. Each row of the stoichiometric matrix represents one unique compound, and each column represents one reaction. The matrix values are stoichiometric coefficients of the participating metabolites in the reaction. When the objective function is appropriately set, fluxes throughout the network can be determined by linear programming to determine the maximum or minimum of the assigned objective function [11]. The strength of a GEM is its simplicity and ability to predict the flux distribution after metabolic perturbation by genetic modifications or environmental changes. GEM has been applied in a wide range of studies of microorganisms and humans [9,12,13].

In general, a GEM contains all possible reactions that organisms of interest can process. This means that some reactions may not occur in specific cell types or under specific conditions. Consequently, integration of omics data representing gene expression, protein expression, or metabolite accumulation levels enables recapitulation of the metabolic pathway in a specific cell type under specific conditions [14]. Particularly, modeling of multicellular organisms such as humans and plants requires the integration of omics data to transform a global GEM into a context-specific GEM [12], which is a subset of the global GEM in which inactive reactions are removed while maintaining metabolic functions in the extracted model. Thus, tailoring the comprehensive GEMs into context- specific networks improves in silico prediction to better representation of the actual metabolism of a cell or tissue [14].

In this study, we established a pipeline for generating context-specific GEMs to understand the metabolic changes in plants under specific conditions. We performed an in silico investigation of metabolic adaptation of *Arabidopsis thaliana* in response to water deficit. We used publicly available transcriptome data to tailor the *Arabidopsis* GEM and reconstructed context-specific metabolic network models. The context-specific GEMs were used to estimate the flux distribution under drought stress and well-watered control conditions. The biomass production rate, which is considered as an important agricultural trait, was predicted and compared to the actual leaf biomass production rate. By comparative analysis of the flux distribution under drought and control conditions, we identified reactions and metabolites associated with drought adaptation. Finally, we discussed the limitations of this method and provided considerations for further applications.

## 2. Results

### 2.1. Reconstruction and Examination of Context-Specific GEMs

#### 2.1.1. Reconstruction of Context-Specific GEMs

AraGEM, a genome-scale metabolic network model of *A. thaliana* [15], was used as a global model in this study (Figure 1a). The model contained 1601 reactions, 1737 metabolites, and 4833 gene-enzyme reaction-association entries and was compartmentalized into the cytosol, mitochondrion, plastid, and peroxisome. The reactions were grouped into 250 metabolic subsystems according to their metabolic functions. AraGEM covers mainly primary metabolism and contains a part of secondary (plant-specialized) metabolism. First, we removed all preset constraints in the model, creating a no-constraint model as the global model before applying the tailoring algorithm (Appendix A). The transcriptome data of *A. thaliana* rosette leaves under progressive drought stress [16] were retrieved from the Gene Expression Omnibus [17,18].

The data set consisted of time-series data from days 1–13 under drought stress and control (well-watered) conditions. The average expression value of four replicates was used as the gene expression value. We selected the Gene Inactivity Moderated by Metabolism and Expression (GIMME) algorithm [19] as a tailoring tool to reconstruct context-specific GEMs. Briefly, GIMME determines active and inactive reactions according to the weights of gene expression and the model’s objective function. In this study, we used the biomass production rate as an objective function (Appendix A). The algorithm added weights according to the quantitative gene expression values. The reaction weights were divided into inactive and active reactions by setting a gene expression threshold. We set the gene expression threshold so that the time-dependent change in the biomass production rate was most consistent with the rate of rosette leaf fresh weight increase. GIMME minimized the usage of low-expression reaction while maintaining the objective function above a certain value. GIMME tailored the global GEM, resulting in 26 context-specific GEMs (13 time points × 2 conditions) (Appendix A). The sizes of the GEMs varied from 1133 to 1184 reactions (Figure 2a) and contained 1322–1358 metabolites (Figure 2b). The number of reactions present only in the drought or control models varied depending on the day of treatment. Drought and control models were similar at the beginning of treatment and showed a larger difference in the number of the reactions in later stages. From days 10 to 13, 22–54 reactions were present only in the drought models (Figure 2a).

In AraGEM, the reactions are grouped into metabolic subsystems according to metabolic pathways in which they are involved. Concerning the subsystem “flavonoid biosynthesis”, more reactions remained in drought GEMs than in control GEMs at the later stage of treatment (Figure 2c). This is consistent with the result of gene ontology analysis and metabolome analysis performed by Bechtold et al. [16], whose transcriptome data were used to tailor the context-specific GEMs. No other subsystem showed a marked difference between drought and control GEMs (Appendix A). When examining the metabolites present only in drought GEMs, we found large numbers of flavonoids, terpenoids, and sugars in later stages (Figure 2d). These metabolites are known as scavengers of reactive oxygen species or osmolytes [5,7,20]. The results are consistent with the actual metabolome data reported by Bechtold et al. [16].

#### 2.1.2. Model Examination Based on Occurrence Percentage of Metabolites

To examine whether the reconstructed context-specific GEMs represented actual metabolic states, we defined the occurrence percentage of metabolites. Among the metabolites in the global GEM, 66 metabolites were detected by metabolite profiling in Bechtold et al. [16]. They were primary metabolites in central carbon metabolism, amino acids, sugars, and a small number of secondary metabolites. The occurrence percentage was defined as the percentage of metabolites in a context-specific GEM among 66 metabolites. The occurrence percentage was around 90%, indicating that the context-specific GEMs had the ability to represent actual plant metabolic states (Figure 3). However, several sugars and sugar phosphates were not included in the context-specific GEMs, indicating that the effect of drought stress on these metabolites cannot be predicted.

### 2.2. Biomass Production Rate

#### 2.2.1. Comparison of Estimated and Actual Biomass Production Rate

We reconstructed the context-specific GEMs so that the time-dependent change in the biomass production rate, the objective function to be maximized (Figure 1b), was consistent with the actual value. As the biomass production rate was not experimentally measured by Bechtold et al. [16], we calculated the rate of fresh weight increase in rosette leaves by using the data reported in the study (Figure 4a; see the Material and Methods section). The rates of fresh weight increase were almost the same between drought and control until day 10, and the difference between drought and control became distinct from days 11 to 13. On the other hand, when the gene expression threshold was appropriately set, the simulated biomass production rates showed similar time-dependent patterns (Figure 4b). This result indicates that context-specific GEMs can properly estimate physiological properties (biomass production rate in this case) in plants under drought stress.

#### 2.2.2. Identification of the Reaction Involved in Increasing Biomass Production Rate

We further investigated which reactions are important for metabolic adaptation in terms of biomass production. First, we identified reactions that accelerated the biomass production rate in control GEMs on days 12 and 13 (Figure 4b). To this end, each single reaction was systematically removed from the day-13 control GEM, after which the biomass production rate was calculated. When reaction ID R00243_c, which corresponds to glutamate dehydrogenase (GDH, EC: 1.4.1.2, 1.4.1.3) in the cytosol, was removed, the biomass production rate was reduced (Figure 5). We further examined the presence or absence of the GDH reaction in other GEMs and found that this reaction was absent from all GEMs except for in control GEMs at days 12 and 13 (Appendix A). To confirm the importance of this reaction, we added the GDH reaction into the day-13 drought GEM and performed FBA. We found that the biomass production rate increased to a level comparable to that in the day-13 control GEM (Figure 5). We also added this reaction to the day-11 control GEM and observed a similar increase in the biomass production rate (Figure 5).

Transgenic tobacco and maize lines overexpressing *Escherichia coli* gdhA were reported to increase their biomass production by enhancing drought tolerance [21,22]. Improved drought resistance was also observed in field experiments [22]. These reports support the ability of this pipeline to identify reactions involved in drought tolerance (Appendix A).

### 2.3. Changes in Flux Distribution Under Drought Stress

We investigated the difference in the flux distribution between drought and control GEMs on each day by flux comparative analysis. We considered that, if the flux size of a reaction is larger under drought conditions than in controls, the reaction was more active under drought conditions. Next, we calculated the fold-change in the flux size (drought/control) (Figure 6a) and counted the reactions showing fold-change values greater than two or less than one-half (Figure 6b). The largest number of reactions with increased or decreased flux size was observed on day 13. This result is consistent with the observation that *Arabidopsis* plants under progressive drought stress showed the greatest physiological changes at day 13 [16]. For instance, the relative leaf water contents were maintained from days 1 to 12 and then began to decline at day 13 [16].

Further, cluster analysis of these reactions was performed (Figure 6c) and metabolic changes during progressive drought stress were summarized (Figure 6d). The reactions were classified into 26 clusters according to their time-dependent patterns. Among them, cluster 3 was notable, as it consisted of reactions that were highly active in later stages, particularly on days 12 and 13. This cluster included the peroxisomal reactions: serine-glyoxylate aminotransferase (R00588_x), hydroxypyruvate reductase (R01388_x), isocitrate hydro-lyase (R01900_x), and citrate hydro-lyase (R01325_x), which are involved in the photorespiration process and glyoxylate cycle. Several transport reactions such as serine transporter (TCX2), glycerate transporter (TCX13), and citrate transporter (TCX14), which are involved in photorespiration, were also included in cluster 3. Nicotinamide adenine dinucleotide (NADH): monodehydroascorbate oxidoreductase (R00095_c and R00095_tmx) is involved in the glutathione- ascorbate cycle. The pyruvate transporter (TCM1) and mitochondrial reactions via pyruvate dehydrogenase (R00209_m) and citrate synthase (R00351_m), which participate in the beginning of the tricarboxylic acid cycle (TCA cycle), were also in this cluster. In contrast, clusters 5–7 consisted of reactions which were less active at the beginning of drought stress and tended to become active in the later stage. Most members participated in pyrimidine metabolism and purine metabolism. Cluster 8 contained the cytosolic glycolysis reactions: beta-D-fructose 1,6-bisphosphate 1-phosphohydrolase (R04780_c) and fructose-bisphosphate aldolase (R01070N_c), which were less active throughout drought treatment. Interestingly, cluster 26 contained the plastidic glycolysis reactions, R04780_p and R01070N_p, which appeared to be active throughout drought treatment. The other reactions in this cluster, R00948_p and R02421_p, are involved in starch and sucrose metabolism. Clusters 9–16 were less active at days 12 and 13. These included the water, CO_2_, and amino acid transporters and reactions associated with the TCA cycle.

### 2.4. Change in Turnover Rate of Metabolites Under Drought Stress

Next, we employed the flux-sum, which is defined as half of the summation of all consumption flux (efflux) and generation flux (influx) related to a specific metabolite under a pseudo-steady state [23,24]. As the size of the flux-sum of each metabolite is closely related to the turnover rate of metabolites, the metabolic state of the system can be examined [23,24]. The flux-sum of each metabolite was calculated using the context-specific GEMs, after which fold-changes (drought/control) were calculated (Figure 7a). The metabolites showed a higher fold-change in later stages of drought stress. Metabolites exhibiting fold-change values of greater than two or less than one-half were classified into 12 clusters, clusters m1–m12 (Figure 7b).

This analysis clarified the metabolic changes from the perspective of metabolites, while cluster analysis of the reactions (Figure 6c) revealed metabolic change in terms of the flux distribution. For example, cluster m10 consisted of metabolites such as hydroxypyruvate and cis-aconitate for which turnover was highly active at later stages. This cluster corresponded to cluster 3 (Figure 6c), which included reactions involved in photorespiration (e.g., hydroxypyruvate reductase; R01388_x) and the glyoxylate cycle (e.g., citrate hydro-lyase; R01325_x). Clusters m2 contained several sugars such as D-glucose, D-fructose, and beta-maltose. These sugars are involved in the reactions of starch and sucrose metabolism clustered in cluster 26. Several nucleosides and nucleotides in clusters m8 and m11 corresponded to clusters 6 and 7, containing reactions involved in pyrimidine and purine metabolism. Interestingly, the metabolites involved in the reactions in clusters 8 and 26, e.g., fructose 1,6-bisphosphate and glyceraldehyde 3-phosphate, did not appear in the metabolite clusters (Figure 7b). This was because the same reactions involving these metabolites were active in the cytosol under control conditions; the plastids under drought stress showed similar turnover rates under both conditions.

As distinct differences were observed at day 13, the metabolic change on day 13 was summarized in a pathway map (Figure 8). Several reactions associated with photorespiration, pyruvate oxidation, plastidic glycolysis, and starch and sucrose metabolism were active under drought conditions. These included peroxisomal reactions (serine-glyoxylate aminotransferase (R00588_x), hydroxypyruvate reductase (R01388_x), isocitrate hydro-lyase (R01900_x), and citrate hydro-lyase (R01325_x)), plastidic reactions (alpha-D-glucose 6-phosphate ketol-isomerase (R02740_p), beta-D-fructose 1,6-bisphosphate 1-phosphohydrolase (R04780_p), fructose- bisphosphate aldolase (R01070N_p), ADP-glucose pyrophosphorylase (R00948_p), starch synthase (R02421_p), glucan phosphorylase (R02111_p), 1,4-alpha-D-glucan maltohydrolase (R02112N_p), and 4-alpha-glucanotransferase (R05196N_p)), and mitochondrion reactions (citrate hydro-lyase (R01325_x) and pyruvate dehydrogenase (R00209_m)). In contrast, the reactions involved in cytosolic glycolysis (alpha-D-Glucose 6-phosphate ketol-isomerase (R02740_c), beta-D-fructose 1,6-bisphosphate 1-phosphohydrolase (R04780_c), and fructose-bisphosphate aldolase (R01070N_c)) were active in the control. In addition, several transport reactions via transporters such as pyruvate transporter (TCM1), serine transporter (TCX2), glycerate transporter (TCX13), and citrate transporter (TCX14), which are involved in cellular respiration and photorespiration, were active under drought conditions.

Some of the abovementioned reactions were reported to be involved in the drought stress response, validating our analytical pipeline for identifying significant reactions related to the drought stress response (Appendix A). Reaction R01070_p corresponds to fructose-bisphosphate aldolase (EC: 4.1.2.13) in the plastid, and the genes encoding this enzyme were highly expressed under abiotic stresses including drought in several plants such as *Arabidopsis*, wheat (*Triticum aestivum* L.), and shoreline purslane mangrove (*Sesuvium portulacastrum*) [25,26,27]. Serine-glyoxylate aminotransferase in the peroxisome (EC: 2.6.1.45; R00588_x) plays an important role in photorespiration during drought stress in barley [28,29]. Hydroxypyruvate reductase in the peroxisome (EC: 1.1.1.81; R01388_x) is involved in the drought stress response, as a mutation in the hydroxypyruvate reductase 1 gene of *Arabidopsis* enhanced the susceptibility to drought stress [30].

## 3. Discussion

We aimed to establish a systematic pipeline to gain insight into metabolic adaptation to environmental changes. The pipeline integrating genome-scale model and omics data allowed us to capture the reactions and metabolites that may enhance drought stress tolerance without prior knowledge. As an example, we clarified that the reaction R00243_c (glutamate dehydrogenase), which was lacking in drought GEMs, is an important reaction affecting the biomass production rate. Previous physiological and molecular genetic studies indicated the significance of glutamate dehydrogenase in drought tolerance [21,22]. However, transcriptome analysis did not suggest the importance of this gene because the genes [16] encoding this enzyme were not induced under drought stress (Appendix A). Further, several reactions in context-specific GEMs (e.g., TCX2, TCX13, TCP1, and TCM1) were shown to be active under drought stress in this study. These reactions lack gene-enzyme reaction associations because the responsible genes have not been identified. This indicates the advantage of using this model to identify candidate active metabolic reactions compared to using only transcriptome data.

In this study, we tailored context-specific GEMs by using the established tailoring algorithms. The tailoring algorithms were categorized into three families: GIMME-like, Integrative Metabolic Analysis Tool (iMAT)-like, and The Model Building Algorithm (MBA)-like families [31]. Among these, we avoided using algorithms in the MBA-like family, which provide only context-specific model reconstruction without the flux distribution. Therefore, three algorithms, GIMME, iMAT, and INIT (Integrative Network Inference for Tissues), were tested in this study. We found that GIMME-tailored models extracted a larger number of reactions and metabolites compared to the other algorithms. The maximum number of reactions in the GEMs tailored by GIMME, iMAT, and INIT were 1184, 576, and 557, respectively. GIMME allowed the model to capture many possible reactions, which is also beneficial for further analysis. Thus, we reconstructed the context-specific model using this algorithm.

To perform in silico analysis using this pipeline, two points should be carefully considered. The first point is setting the threshold of the expression value of transcriptome data. The threshold setting has a major effect on the content of the tailoring model [14]. In this study, several thresholds were examined (Appendix A). The less restrictive threshold did not provide a clear difference between the control and drought conditions, whereas a more restrictive threshold lost reactions that may be active in the plant system. We chose a threshold that gave the time-dependent pattern of the biomass production rate which was most consistent with the rate of increase in the rosette leaf weight (Figure 4a,b). Thus, it is recommended to set a suitable objective function for a plant trait exhibiting an apparent difference between control and target conditions. The second point is the robustness and capabilities of the global model. Although AraGEM has been assessed for its robustness and capabilities in numerous in silico simulations, secondary metabolism is largely neglected and it is recommended to manually add secondary metabolites to the biomass constituting equation [32]. Although our drought-specific GEMs contained more reactions related to flavonoid biosynthesis than control models (Figure 2b), the flux was not distributed to these reactions in our simulation because flavonoids were not included in the biomass production reaction (Appendix A).

The pipeline proposed in this study can be applied in studies aimed at understanding metabolic adaptation to stresses other than drought. Plant GEMs have been reported for several species, including rice, maize, and soybean [32]. The pipeline is also applicable to genome-scale metabolic models of other organisms.

## 4. Materials and Methods

### 4.1. Genome-Scale Metabolic Model

AraGEM, a genome-scale metabolic model of *A. thaliana,* was downloaded as an sbml file from [15]. The downloaded model contained preset constraints, such as photosynthesis condition of the leaf cell, which were removed prior to use. Thus, the lower and upper bound constraints were set to 0 and infinite, respectively, for the irreversible reactions, whereas both bound constraints were set to infinite for reversible reactions. Kyoto encyclopedia of genes and genomes (KEGG) IDs were retrieved via KEGG Application programming interface (API), a Representational state transfer (REST) application programming interface, to the KEGG database using BioServices python framework version 1.6.0 [33,34], whereas the missing compound IDs were reiterated with manual curation. The no-constraint model was saved as a new sbml file for use in later steps.

### 4.2. Transcriptome Data Processing

Microarray data were retrieved from Gene Expression Omnibus under accession number GSE65046 [16] using package GEOquery version 2.52.0 in R [35]. Briefly, signal intensities of microarray data were normalized within array by Lowess normalization, and then, variation due to arrays and dyes was removed by a random effects model and averaged in log space using a modified version of R/MAANOVA (MicroArray ANalysis of VAriance) [16,36]. Using the getGEO function, the deposited normalized expression values of 32,501 feature probes of 108 samples were downloaded. The gene expression values were calculated from a mean of replicated samples under individual conditions. The gene expression values were subsequently used to tailor context-specific GEMs.

### 4.3. Calculation of the Rate of Fresh Weight Increase

The rate of fresh weight increase in rosette leaves was used to represent the actual biomass production rate. From the experimental data reported by Bechtold et al. [16], the fresh weights of rosette leaves were used for Equation (1).
(1)$$\mathrm{Rate}\;\mathrm{of}\;\mathrm{fresh}\;\mathrm{weight}\;\mathrm{increase}=\frac{\Delta \mathrm{rosette}\;\mathrm{leaf}\;\mathrm{fresh}\;\mathrm{weight}}{\Delta t}$$

### 4.4. Flux Balance Analysis (FBA)

FBA was performed on context-specific GEMs to simulate the maximum possible flux of the feasible flux distributions [11]. The objective function (*Z*) was the biomass production rate ($v_{biomass}$). Gurobi Optimizer v8 (Academic version) was used to solve the given objective function by linear programming in Equation (2).
(2)$$\mathrm{Maximize}\;Z=c^{T}\times v_{biomass}\;\mathrm{subject}\;\mathrm{to}\;S\times v=0\;l<v<u$$
where $c\in {ℜ}^{n}$ is a vector of reaction coefficient contributing to the objective function; $S\in {ℜ}^{m\times n}$ is a matrix of the stoichiometric coefficient for *m* metabolites and *n* reactions; and $v\in {ℜ}^{n}$ is a vector of flux distribution of all reactions that maximizes the objective function and stands between lower and upper bounds, *l* and *u*, respectively. To avoid the possibility of multiple optima during optimization which results in different flux distribution even with the same optimal objective flux, the objective (*Z*) was regularized by subtraction with a strictly concave function equation [37]. The minimization of the squared Euclidean norm of internal flux was performed by setting an optimization parameter called “minNorm” to be 1 × 10^−6^. The FBA was performed using COBRA toolbox version 3.0 in MATLAB (version 9.5-R2018b) [37].

### 4.5. Reconstruction of Context-Specific GEMs Using GIMME

Reconstruction of context-specific GEMs was performed using Gene Inactivity Moderated by Metabolism and Expression (GIMME) [19]. Briefly, GIMME removed inactive reactions below a specified threshold of low expression genes and then reinserted reactions required for the objective function to produce at or above a certain level. First, the algorithm found the maximum possible flux through the objective function (e.g., biomass production rate in this study) by performing FBA in Equation (1) and used the flux for the reaction bound in the following step. Second, the algorithm identified the active reactions by minimizing inactive reactions according to the gene expression-weight coefficient by following linear programming in Equation (3).
(3)$$\mathrm{minimize}\;Z=\sum^{}x\times \left|v\right|\;\mathrm{subject}\;\mathrm{to}\;S\times v=0\;a<v<b\;\mathrm{where}\;x=\{x_{threshold}-x_{n}\;\mathrm{where}\;x_{threshold}>x_{n},\;0\;\mathrm{otherwise}\;\mathrm{for}\;\mathrm{all}\;n\}$$
where $x\in {ℜ}^{n}$ is a vector of gene expression-weight reaction coefficients of *n* reactions calculated from a setting threshold value and gene expression values mapped to each reaction, $x_{threshold}$ and $x_{n}$, respectively. The expression values above the threshold were excluded from the minimization ($x_{n}=0$ where $x_{threshold}<x_{n}$). $S\in {ℜ}^{m\times n}$ is a stoichiometric matrix of *m* metabolites and *n* reactions, $v\in {ℜ}^{n}$ is a flux vector describing flow through all reactions, and *a* and *b* were lower and upper bounds, respectively. The maximal value of the objective value was set to 90%. The lower bounds, corresponding to the objective function (*Z*), were set to the maximal value, whereas other bounds were set as the original bounds.

The expression threshold was set to be the same on all days and under all conditions. The expression thresholds ranged from 0.7 to 0.9 (0.01 for intervals) were tested (Appendix A). Then, we decided to set the expression threshold at quantile 0.83, at which the time-dependent pattern of biomass production rate was most similar to that of the fresh weight increase rate. The above procedures were performed using COBRA toolbox version 3.0 in MATLAB (version 9.5-R2018b) [37]. The example script is provided as Appendix A.

### 4.6. Occurrence Percentage

The occurrence percentage of metabolites in context-specific GEMs was calculated for each GEM by comparison with the metabolites for which contents were reported in Reference [16]. If a metabolite was not present in the global AraGEM, that metabolite was excluded before calculation. The occurrence percentages were calculated among 66 metabolites using the following Equation (4).
(4)$$\mathrm{Occurrence}\;\mathrm{percentage}\;\left(\%\right)=\frac{\mathrm{The}\;\mathrm{number}\;\mathrm{of}\;\mathrm{present}\;\mathrm{metabolites}\;\mathrm{in}\;\mathrm{GEM}}{\mathrm{The}\;\mathrm{number}\;\mathrm{of}\;\mathrm{reported}\;\mathrm{metabolites}}$$

### 4.7. Single Reaction Deletion

Each single reaction in the day-13 control GEM was deleted individually using the singleRxnDeletion function from COBRA toolbox version 3.0 in MATLAB (version 9.5-R2018b) [37]. Next, the biomass production rate of each single reaction-deleted model was calculated and compared to that calculated using day-13 drought GEM. If a deleted model gave the same value, this reaction was selected for insertion to the day-13 drought GEM. FBA was then performed, and the biomass production rates before and after insertion were compared.

### 4.8. Flux-Sum

Flux-sum is defined as one-half of the summation of all incoming fluxes (influx) and outgoing fluxes (efflux) concerning individual metabolite ($x_{i}$) in Equation (5) [23]:(5)$$v_{x_{i}}=\frac{1}{2}\underset{j}{\sum }\left|s_{ij}v_{j}\right|$$
where $s_{ij}$ refers to the stoichiometric coefficient of metabolite *i* participating in reactions (*j*) and $v_{j}$ is the flux of reactions (*j*).

### 4.9. Comparative Analysis of Flux and Flux-Sum

#### 4.9.1. Fold-Change in Flux

All reactions present in each context-specific GEM were listed to create a matrix of simulated fluxes, $F\in {ℜ}^{m\times n}$, where *m* is the number of context-specific GEMs and *n* is the number of reactions. The missing values in the matrix were filled with one-tenth of the minimum value of flux sizes. The fold-change in flux size under drought to control conditions was calculated in Equation (6) and transformed into logarithm to base 2.
(6)$$\mathrm{Fold}\;\mathrm{change}=\frac{F_{drought_{n}}}{F_{control_{n}}}$$

#### 4.9.2. Fold-Change in Flux-Sum

All metabolites present in each context-specific GEM were listed to create a matrix of flux-sum values, $Fs\in {ℜ}^{m\times n}$, where *m* is the number of context-specific GEMs and *n* is the number of metabolites. The missing values in the matrix were filled with one-tenth of the minimum value of flux-sum values. The fold-change in flux-sum under drought to control conditions was calculated in Equation (7) and transformed into logarithm of base 2.
(7)$$\mathrm{Fold}\;\mathrm{change}=\frac{Fs_{drought_{n}}}{Fs_{control_{n}}}$$

#### 4.9.3. Cluster Analysis

Cluster analysis was performed using agglomerative clustering using a correlation metric with average linkage. Reactions showing a correlation greater than 0.5 were grouped into clusters and are shown in different colors. Heatmap representation was performed using python library SciPy and library Seaborn [38,39].

## 5. Conclusions

In this study, context-specific GEMs were reconstructed using transcriptome data under drought stress. In silico single reaction deletion experiments revealed the reactions important for increasing the biomass production rate. Comparative analysis between drought and control conditions clarified the candidate reactions associated with the drought stress response, which can be utilized, after further biological validation, to improve the drought tolerance of crops. This strategy can be applied to other stresses to gain a better understanding of metabolic pathways affecting the stress response involved in growth and yield, which will help improve future agricultural strategies.

## Figures and Tables

**Figure 1 metabolites-10-00159-f001:**
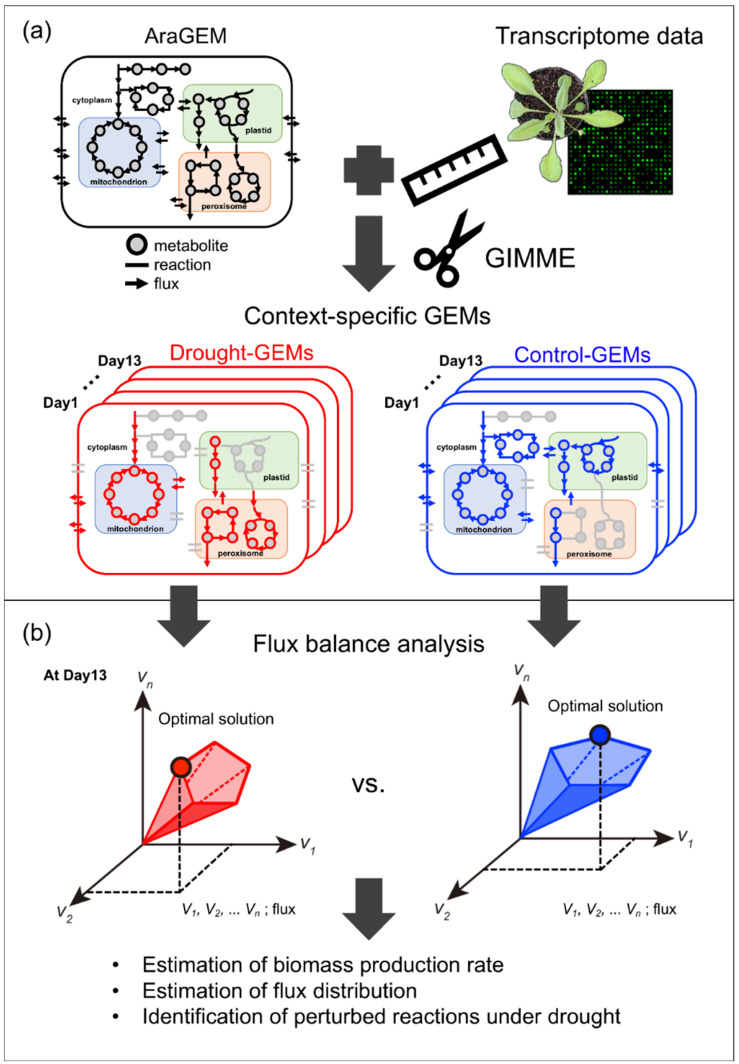
Outline of this study for extracting information for the metabolic network by reconstructing context-specific genome-scale metabolic models (GEMs): (**a**) Transcriptome data were used to tailor the global GEM of *A. thaliana* and to obtain context-specific GEMs under progressive drought treatment (days 1–13). (**b**) Using flux balance analysis (FBA), the biomass production rate and flux distribution were estimated. Comparative analysis of the flux distribution between drought and control conditions was performed to identify candidate reactions associated with adaptation to drought. Drought and control GEMs are colored in red and blue, respectively. The eliminated reactions and metabolites are shown in grey. Examples of optimal solutions for flux distribution that lies on the border of the feasible space are represented by red and blue dots for drought and control GEMs, respectively.

**Figure 2 metabolites-10-00159-f002:**
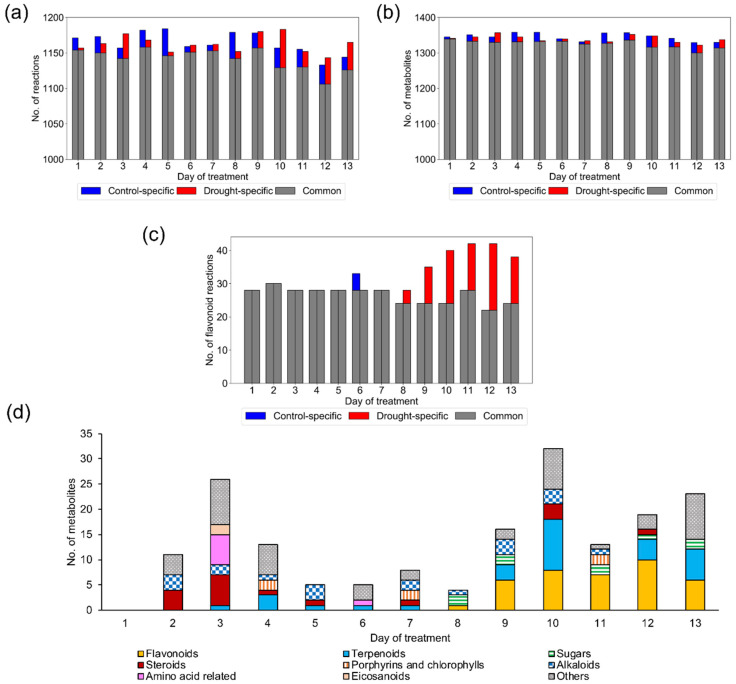
Features of context-specific GEMs: (**a**) Total number of reactions in each GEM, (**b**) total number of metabolites in each GEM, and (**c**) number of reactions in the “flavonoid biosynthesis” subsystem in each GEM. In Figure 2a–c, the left and right bars in a day represent the number in control and drought GEMs, respectively. The control-specific, drought-specific, and common reactions are displayed in blue, red, and grey, respectively. (**d**) Number of metabolites present only in drought GEMs: Metabolites are categorized based on KEGG BRITE database (Kyoto encyclopedia of genes and genomes - functional hierarchies of biological entities: https://www.genome.jp/kegg/brite.html) and shown in different colors.

**Figure 3 metabolites-10-00159-f003:**
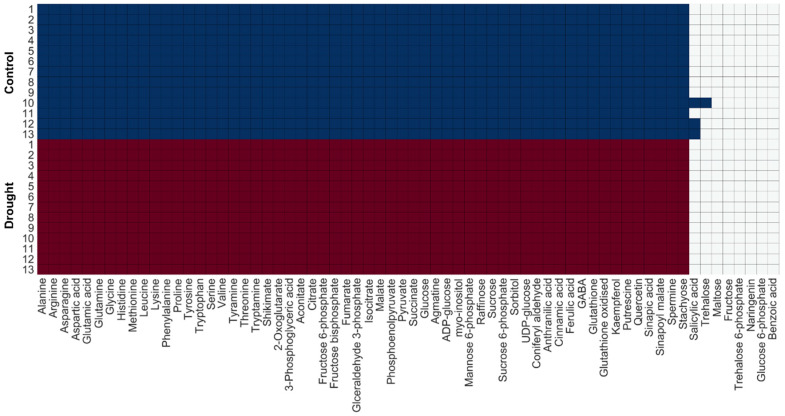
The list of 66 metabolites indicating occurrence percentage of each GEM: Blue and red represent metabolites present in control and drought GEMs, respectively.

**Figure 4 metabolites-10-00159-f004:**
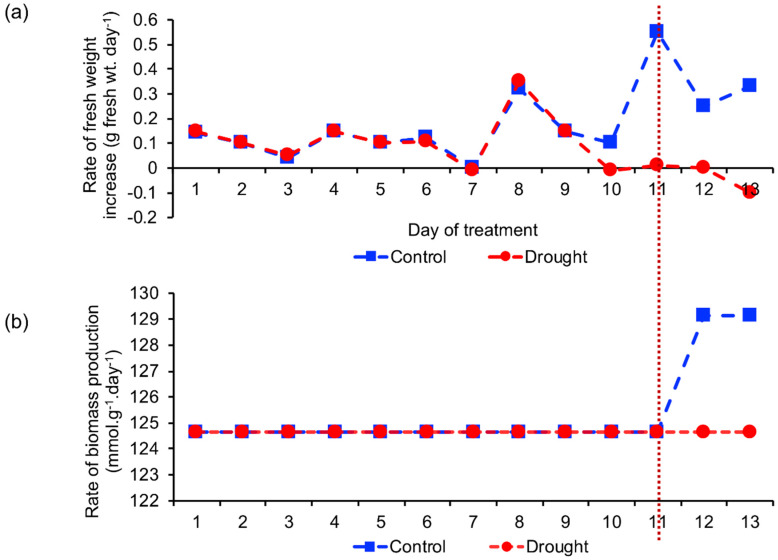
Comparison of estimated and actual biomass production rates: (**a**) Rate of fresh weight increase in rosette leaves in biological experiment [16] and (**b**) rate of biomass production calculated using the context-specific GEMs. Blue, control; red, drought.

**Figure 5 metabolites-10-00159-f005:**
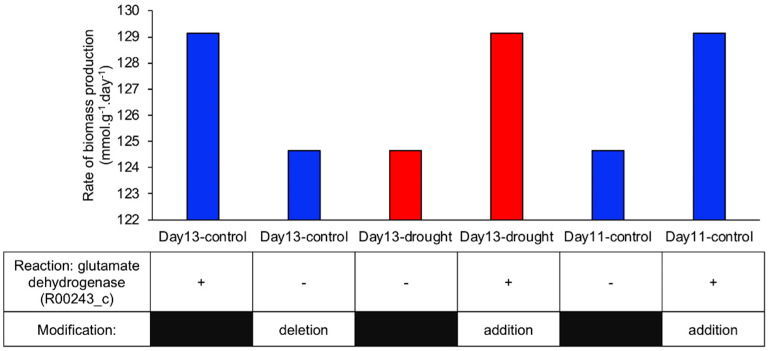
Importance of glutamate dehydrogenase (GDH) reaction for the increased biomass production rate in GEMs: Blue and red bars represent the biomass production rate calculated in the control and drought GEMs, respectively.

**Figure 6 metabolites-10-00159-f006:**
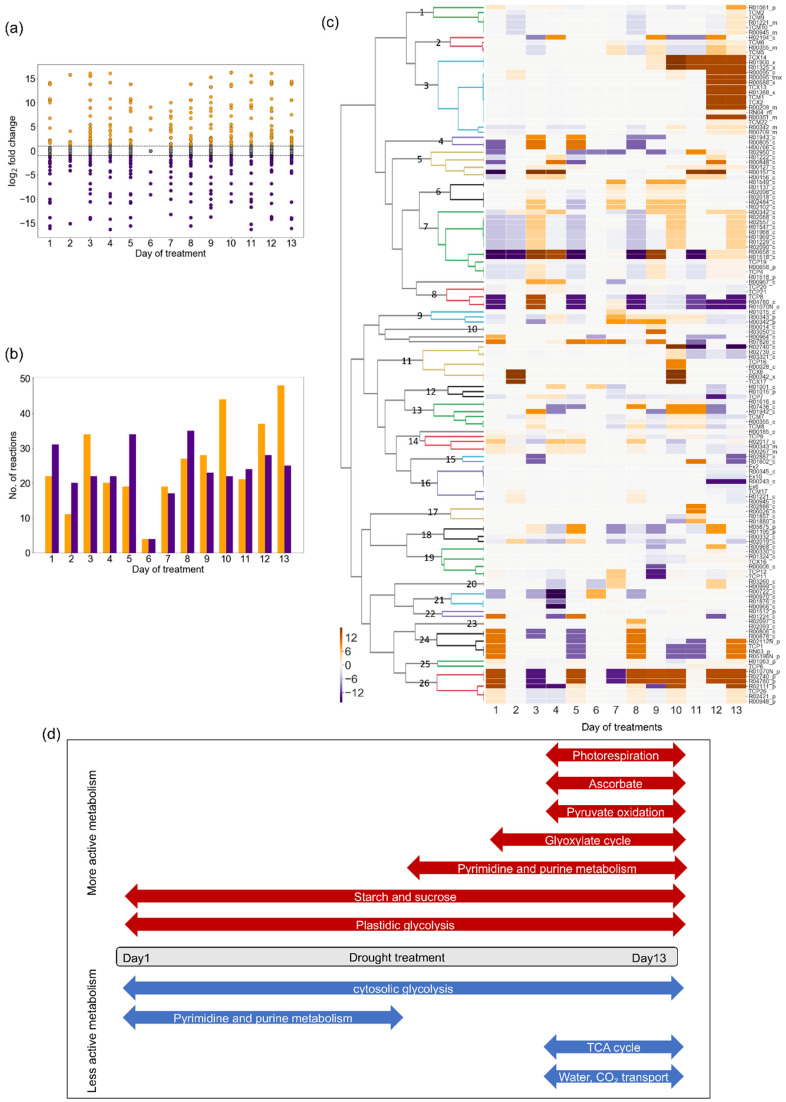
Changes in flux distribution: In the figures, fold-change (drought/control) value was transformed into logarithm to base 2 for clarity. (**a**) Distribution of log_2_(fold-change) values and (**b**) number of reactions showing fold-change values greater than two or less than half (namely, |log_2_(fold-change)| > 1). In Figure 6a,b, orange and purple indicate increased and decreased flux under drought, respectively. (**c**) Clustering analysis of reactions showing fold-change values greater than two or less than half at least one time point. (**d**) Summary of metabolic change during progressive drought stress.

**Figure 7 metabolites-10-00159-f007:**
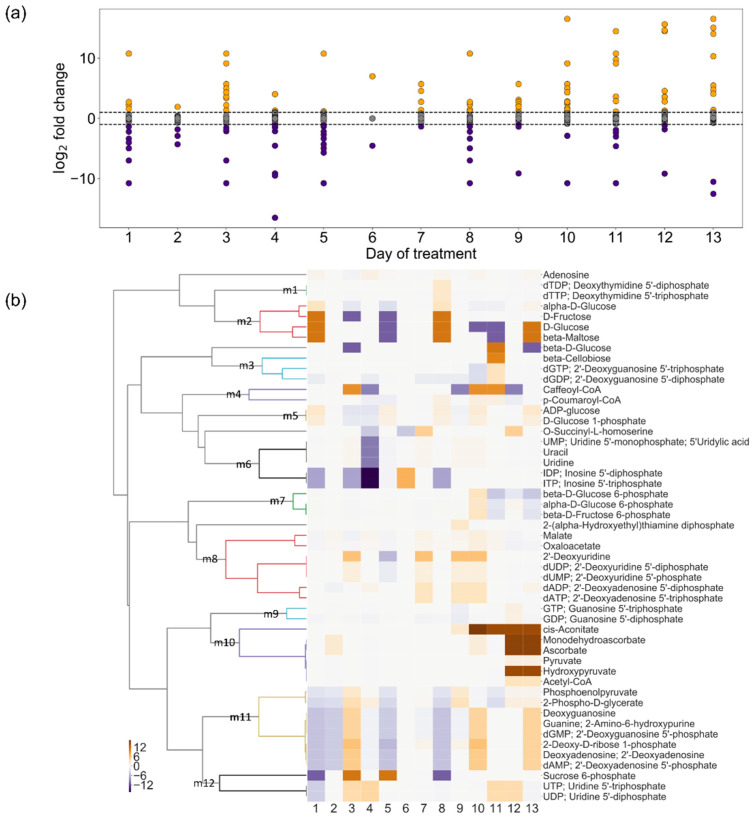
Changes in flux-sum: In the figures, the fold-change (drought/control) value was transformed into logarithm to base 2 for clarity. (**a**) Distribution of log_2_(fold-change) values: Orange and purple dots represent the metabolites with fold-change values greater than two or less than one- half, respectively. (**b**) Cluster analysis of the metabolites with fold-change values greater than two or less than half for at least one time point.

**Figure 8 metabolites-10-00159-f008:**
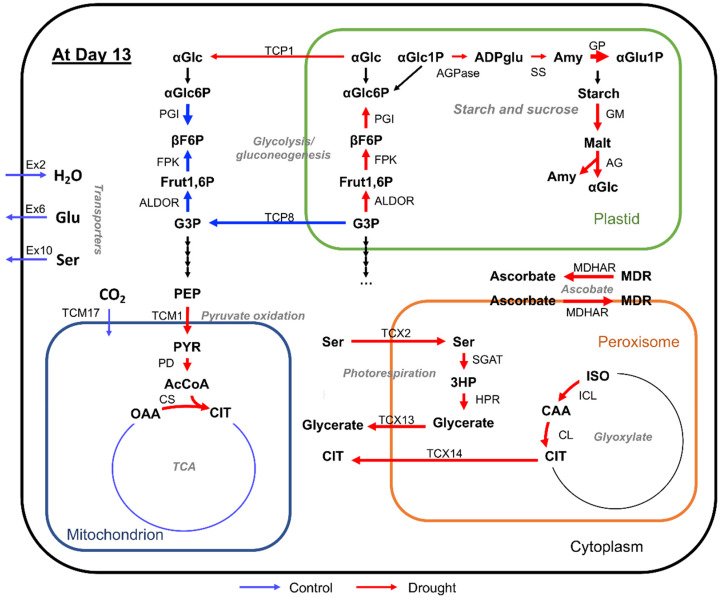
Metabolic pathway showing the reactions involved in drought stress response at day 13: Red and blue arrows represent active reactions in drought and control, respectively. The thickness of arrows represents fold-change values. Metabolites: αGlc, alpha-D-glucose; αGlc6P, alpha-D-glucose 6-phosphate; βF6P, beta-D-fructose 6-phosphate; Fruc1,6P, fructose 1,6-bisphosphate; G3P, glyceraldehyde 3-phosphate; PEP, phosphoenolpyruvate; PYR, pyruvate; AcCoA, acetyl-CoA; OAA, oxaloacetate; CIT, citrate; Ser, serine; 3HP, 3-hydroxypyruvate; CAA, *cis*-aconitate; ISO, isocitrate; MDR, monodehydroascorbate; Glu, glutamate; αGlc1P, alpha-D-glucose 1-phosphate; ADPglu, ADP-glucose; Amy, amylose; Mal, maltose. Reactions: PGI, alpha-D-Glucose 6-phosphate ketol-isomerase (R02740_c and R02470_p); FPK, beta-D-Fructose 1,6-bisphosphate 1-phosphohydrolase (R04780_c and R04780_p); ALDOA, fructose-bisphosphate aldolase (R01070N_c and R01070N_p); SGAT, serine- glyoxylate aminotransferase (R00588_x); HPR, hydroxypyruvate reductase (R01388_x); ICL, isocitrate hydro-lyase (R01900_x); CL, citrate hydro-lyase (R01325_x); PD, pyruvate dehydrogenase (R00209_m); CS, citrate synthase (R00351_m); MDHAR, NADH: monodehydroascorbate oxidoreductase (R00095_c and R00095_tmx); AGPase, ADP-glucose pyrophosphorylase (R00948_p); SS, starch synthase (R02421_p); GP, glucan phosphorylase (R02111_p); GM, 1,4-alpha-D-glucan maltohydrolase (R02112N_p); AG, 4-alpha-glucanotransferase (R05196N_p); Ex2, water transporter; Ex6, glutamate transporter; Ex10, serine transporter; TCP1, glucose translocator between cytoplasm and plastid; TCP8, triose phosphate translocator between cytoplasm and plastid; TCM1, pyruvate translocator between cytoplasm and mitochondrion; TCM17, CO_2_ translocator between cytoplasm and mitochondrion; TCX2, Serine translocator between cytoplasm and peroxisome; TCX13, glycerate translocator between cytoplasm and peroxisome; TCX14, Citrate transporter between cytoplasm and peroxisome.

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
