# Peer review of "Drought Stress Responses in Context-Specific Genome-Scale Metabolic Models of Arabidopsis thaliana"

_metabolites, 2020, doi:10.3390/metabo10040159_

Round 1

Reviewer 1 Report

This paper analyzed the metabolic pathway which responds to drought stress by omics-based in silico analyses. They used data of Bechtold et al plant cell 2016 to analyze the AraGEM (Dal'Molin et al 2010). In this study, the context-specific GEM was reconstructed to identify an important reaction contributing to biomass production and clarified metabolic reaction responses.  The former paper of Bechtold et al plant cell 2016 already found the total of 1815 drought-responsive differentially expressed genes, especially the AGL22 expression influences steady state photosynthetic rates and lifetime water use.  However, the conclusion of this paper is not impressive as they gave some candidate reactions associated with the drought stress response. The suggested the example of reaction R00243_c (glutamate dehydrogenase) without verification, although E.coli GDH could improve the drought tolerance in some plant species. More example reaction should be listed in one table, such as proline, ABA et al with reference or verification.

Minor points

Figure 6, 7, 8, did not mark the control and drought

Author Response

Thank you for your helpful comments. We have answered each of your points. Please see the attachment.

Reviewer 2 Report

Section 2.1.2 is confusing. Can the authors clarify what they define as “occurrence percentage” and how this qualifies as a validation method for genome-scale metabolic models? Are the authors considering all the metabolites in the model(s) in this section? It seems that the occurrence percentage for metabolites is exactly same for both control and drought samples, which the only exception being the control samples at 10, 12, and 13 days? What are these metabolites? What pathways do they belong to and what role do they play?

Can the authors elaborate on how they estimated the increase in biomass production rate for rosette leaf from experimental data (section 2.2.1)? How did the authors conclude that the model properly estimated the physiological properties under stress? What physiological properties they compared between the model and experiment?

AraGEM is a global plant model, which does not consider the tissue-specific metabolic differences. Since the authors are utilizing transcriptomic data from rosette leaves in their model, are they thereby arguing that “filtering out” reactions from the global model using transcriptomic data is enough to develop a leaf-specific model?

The take home message from Figure 4 is not clear. What is seen here is that there is not difference in the biomass production between control and drought-stressed models, and only at day 12 and day 13 the control model shows slightly higher biomass production rate, none of these two observations are coherent to the experimental observation. In fact, there should be a decrease in the biomass for the drought-stressed model. From the transcriptomic data, did the authors consider the gene downregulated in stress condition? Why was there no effect on biomass flux for the stressed model? The constant value of biomass production in stressed model at all time points (11 of which are exactly same as the control) indicate that the model can redistribute fluxes to satisfy the same objective value. This point to an even more serious issue of degeneracy in flux distribution, especially when FBA is applied to generate results.

Section 2.2.2: Why is there one single reaction responsible for the increase in biomass in the control model at day 12 and 13? If glutamate dehydrogenase is only active in those time points, then the decrease in biomass should also be observed in all those other time points, which is observed in the experimental data. So it is not clear what the conclusion about drought tolerance is.

The authors should use reaction/protein names instead of reaction IDs throughout the manuscript. Reaction IDs do not convey a lot of information, and they vary across model reconstructions among different research group. Please do this for the figures as well (for example, Figure 8).

Why wasn’t the more updated Arabidopsis GEM (constructed by Cheung et al.) used?

How was the transcription data preprocessed before incorporating it into the GIMME formulation?

Line 99, Cite Gene Expression Omnibus.

Line 111: is it “content-specific GEM” or “context-specific GEM”?

Author Response

Thank you for your valuable suggestion. We have answered each of your points. Please see the attachment.

Reviewer 3 Report

The authors analyzed a genome-scale metabolic network model of Arabidopsis thaliana under two different conditions (drought and control). Previously published transcriptome data was used to create context specific network models for each condition and day where data was available. The authors interpret their computational results by contrasting inferred metabolic network and flux differences between the considered conditions.

The contextualization of the metabolic network models follows an established procedure called GIMME. A key tuning parameter in this procedure is the threshold for the expression level from which a reaction should be considered active. Although the authors discuss the choice of this tuning parameter a bit, I think this point requires further analysis and evidence. Currently it seems that the authors are simply tuning the threshold to obtain a qualitative match between the experimental leave biomass growth and the biomass production rate in the model. However, it is unclear whether the same value for the threshold was used for all days and both conditions. Also, a sensitivity analysis of how changes in the threshold affect the results is lacking. Without such further details, it is hard to convince readers that the proposed network contextualization reliably represents what is going on physiologically in the real system.

The authors try to validate their contextualization with published metabolite data for drought and control conditions. Unfortunately the validation is not very convincing. The validation is only qualitative because only lists of metabolites are compared, not any metabolite levels. It is unclear which dataset regarding the metabolite list in [16] is actually used, since the original publications uses different experimental technique. It as also not clear what the reported "occurence percentages" are actually validating. Apparently a somewhat high (though not very high) percentage of experimentally studied metabolites are also present as reactants in the contextualized metabolic network models used in this study. Many of these are core plant metabolites. I fail to see how that can be interpreted as a validation of the contextualized networks.

Because of the short-comings in network construction (threshold parameter) and validation, the biological relevance of the results remains unclear. Also the authors in their discussion point out several missing factors in the underlying global network that may be problematic for their study. Would it not be better to address these missing factors before drawing conclusions from premature model analyses?

Author Response

(The authors gave the same response as above.)

Round 2

Reviewer 1 Report

I am happy for this version and support it to be published.

Reviewer 2 Report

The comments/concerns were addressed in this version.